# Nitrates as a Potential N Supply for Microbial Ecosystems in a Hyperarid Mars Analog System

**DOI:** 10.3390/life9040079

**Published:** 2019-10-19

**Authors:** Jianxun Shen, Aubrey L. Zerkle, Eva Stueeken, Mark W. Claire

**Affiliations:** School of Earth and Environmental Sciences and Centre for Exoplanet Science, University of St Andrews, St Andrews KY16 9AL, Scotland, UK; az29@st-andrews.ac.uk (A.L.Z.); ees4@st-andrews.ac.uk (E.S.); mc229@st-andrews.ac.uk (M.W.C.)

**Keywords:** nitrate, Mars, Atacama Desert, sedimentation rates, biomass preservation, extremophiles

## Abstract

Nitrate is common in Mars sediments owing to long-term atmospheric photolysis, oxidation, and potentially, impact shock heating. The Atacama Desert in Chile, which is the driest region on Earth and rich in nitrate deposits, is used as a Mars analog in this study to explore the potential effects of high nitrate levels on growth of extremophilic ecosystems. Seven study sites sampled across an aridity gradient in the Atacama Desert were categorized into 3 clusters—hyperarid, middle, and arid sites—as defined by essential soil physical and chemical properties. Intriguingly, the distribution of nitrate concentrations in the shallow subsurface suggests that the buildup of nitrate is not solely controlled by precipitation. Correlations of nitrate with SiO_2_/Al_2_O_3_ and grain sizes suggest that sedimentation rates may also be important in controlling nitrate distribution. At arid sites receiving more than 10 mm/yr precipitation, rainfall shows a stronger impact on biomass than nitrate does. However, high nitrate to organic carbon ratios are generally beneficial to N assimilation, as evidenced both by soil geochemistry and enriched culturing experiments. This study suggests that even in the absence of precipitation, nitrate levels on a more recent, hyperarid Mars could be sufficiently high to benefit potentially extant Martian microorganisms.

## 1. Introduction

Nitrogen (N) is one of the essential elements in amino acids and nucleic acids, which are fundamental building blocks for terrestrial life. Gaseous nitrogen constitutes only 2.6% of the 6 mbar Martian atmosphere [1], much lower than the 78% of N_2_ in Earth’s 1013 mbar atmosphere, and, therefore, could be considered as limiting to (past or present) Martian life in terms of atmospheric source [2]. Additionally, N_2_ is a fairly inert molecule, and it is uncertain if other biospheres would necessarily have developed the ability to enzymatically convert it into bioavailable forms. However, soluble nitrate salts, which are commonly preserved in recent Martian sediments because of the lack of a leaching mechanism [3], could provide an alternate N source for potential Martian life.

Approximately 4.1 to 3.0 billion years ago (Noachian and early Hesperian Periods) [4,5], the existence of ancient deltas, paleolake basins, and carved outflow channels suggests that Mars was a more humid planet, potentially supporting oceans [6,7], rivers, lakes [8,9], and possibly, rainfall [10,11]. Given the importance of water for biochemical functions, life presumably could have evolved and thrived during that period. If so, during the transition to the current extremely hyperarid planet, Martian life could have faced harsh conditions similar to those experienced by microbial communities in hyperarid environments on Earth. In spite of the absence of detectable rainfall on modern Mars, the percent hydration of soil amorphous materials was discovered to be 5%–9% by ChemCam analyses [12,13]. Viking 1 and 2 rovers also determined the water abundance in Martian soils to be between 1% and 3% [14]. At Gale crater on Mars, Curiosity rover has measured ground-level relative humidity of up to 30%–50% [15], which is higher than the mean relative humidity determined in the hyperarid core of the Atacama Desert [16]. Under such low-water conditions, only extremophilic life that is tolerant of desiccation, irradiation, and high salinity could potentially survive and even reproduce [17]. To achieve this, extremophiles would have needed to develop strategies to take advantage of ambient N sources, such as atmospherically derived nitrate salts.

The Atacama Desert has been used as a terrestrial Mars analog [18] due to its extreme dryness and high UV flux. The Atacama Desert occupies around 105,000 km^2^ of northern Chile, and is enclosed in the rain shadow of two mountain ranges (the Coastal Cordillera and Altiplano/Andes Mountains) [19]. Due to this unique geographic location, combined with the effects of the Humboldt current and the Pacific anticyclone [20], the Atacama has remained dry for the past 150 Myr [21] and extremely dry (with annual precipitation at the low elevation hyperarid core less than 2 mm [22]) for more than 10 Myr [23]. Within the hyperarid core, the Yungay region has been studied the most as a dry limit of life, with the mean annual precipitation as low as 0.7 mm during typical dry years (1994–1998) [20]. In addition, both the high- and low-altitude Atacama Deserts are exposed to extreme surficial UV irradiation as high as ~150 kWh/m^2^ in the UV-A (315–400 nm) and ~5 kWh/m^2^ in the UV-B (280–315 nm), more than 40% greater than the average UV-B intensity in Northern Africa [24]. UV irradiation in these bands can both destroy organic matter [25,26] and photochemically produce oxyanions [27] such as nitrate [18,28], the large reservoir of which is a unique feature of the Atacama Desert among all deserts on Earth [29,30]. Since rainfall is extremely limited in the Atacama Desert, atmospherically produced salts are not solubilized as elsewhere on earth, and can build up to very high levels [31,32]. Some of the most abundant Atacama nitrate minerals include nitratine (commonly called soda niter or Chilean saltpeter), darapskite, and humberstonite [30].

Nitrate was recently detected in mudstone deposits at Gale Crater on Mars at levels of 70 to 1100 parts per million (ppm) by the Mars Science Laboratory [3]. Similar nitrate levels were determined in Martian meteorites, such as EETA79001 [33] and Tissint [34]. These levels are very close to nitrate levels measured in Atacama sediments during previous studies [28,35,36,37]. The large repositories of nitrate in the Atacama Desert are a product of atmospheric photolytic reactions, as indicated by the anomalous Δ^17^O signals in Atacama salts [28,35]. Analogous processes could partially contribute to Martian nitrate deposits [38]. Some Martian nitrate could also originate from carbon dioxide–nitrogen reactions in the ancient Martian atmosphere stimulated by impact shock heating [39,40]. Thermal decomposition triggered by the late heavy bombardment could also have reversely converted Martian nitrate back to atmospheric nitrogen, redistributing the sedimentary nitrate content and balancing the equilibrium between nitrate and atmospheric nitrogen [41,42].

High levels of nitrate that build up under hyperarid conditions, like the Atacama Desert and the recent Martian surface, could potentially provide the dominant source of bioavailable N for local microbial communities in the subsurface. Nitrate is a significant N source for a variety of microorganisms [43]. Nitrate acts not only as a source of N for biomass production, but also an energy source for life in chemotrophic metabolisms. Being an oxidizing agent, nitrate is an important electron acceptor in nature [44]. Nitrate (or nitrite) can be used in chemotrophic metabolisms such as denitrification, dissimilatory reduction of nitrate to ammonium (DNRA), and anaerobic ammonium oxidation [45,46]. Therefore, beyond its universal importance in biomass production, nitrate can further stimulate the growth of microorganisms that utilize these chemotrophic metabolisms. Laboratory experiments have further demonstrated that microorganisms are capable of utilizing surrounding nitrate as the only nitrogenous source for survival [47,48].

The majority of previous investigations into microbial N cycling have focused on marine or terrestrial environments, where liquid water is abundant or non-limiting and nitrate is aqueous or easily solubilized [49,50]. Here, we investigate the effects of nitrate on microbial growth in a hyperarid-arid Mars analog, to investigate how microorganisms survive in a water-limited, oligotrophic natural environment with high levels of nitrate [35]. We combine a number of geochemical and microbiological methods to study how nitrate accumulation as a result of rain limitation and long-term atmospheric deposition might impact the microbial abundance and growth rates in these soils.

## 2. Materials and Methods 

### 2.1. Study Sites

Seven sites were sampled in the Atacama Desert, on a latitudinal gradient from 22° S to 29° S (Figure 1), between the 30 November and 6 December 2017, half a year after an unprecedented rainfall event (centered on the region of Yungay, during 6–7 June of 2017) [36]. From the hyperarid north to the arid south, the sites are María Elena South (MES), Point of No Return Dos (PONR-2), Yungay, Transition Zone 0 (TZ-0), Transition Zone 4 (TZ-4), Transition Zone 5 (TZ-5), and Transition Zone 6 (TZ-6). TZ-0 is located near the north border of our loosely-defined “transition zone” of the Atacama Desert which has combined environmental features of the hyperarid core and the more humid arid sites to the south, which feature rainfalls greater than 15 mm/yr. Vegetation became visible from site TZ-4 during our sampling dates (Figure 2). Dead brush and the presence of small desiccated roots which were not at the site when visited in 2012 imply that a desert bloom (*desierto florido*) [51] may have occurred at TZ-4 prior to our sampling, which may have affected subsurface soluble salt levels.

These sites are situated on different bedrock geology, as shown in Figure 1. In detail, MES soils are primarily fed by Quaternary and Jurassic sedimentary rocks, Mesozoic volcanics, and Mesozoic–Cenozoic intrusives. PONR-2 soils form on Quaternary sedimentary and Mesozoic volcanic rocks. Yungay soils form from Paleozoic–Mesozoic and Mesozoic–Cenozoic intrusives, and Quaternary sedimentary rocks. TZ-0 soils form from Quaternary and Jurassic–Cretaceous sedimentary rocks. TZ-4 soils form on a mixture of Paleozoic–Mesozoic intrusives, Cretaceous–Tertiary volcanics, and Quaternary sedimentary rocks. TZ-5 soils are affected by Quaternary and Cretaceous sedimentary rocks, and Paleozoic–Mesozoic and Mesozoic–Cenozoic intrusives. Finally, TZ-6 soils are affected by Quaternary and Cretaceous sedimentary rocks, as well as Mesozoic–Cenozoic intrusives [52,53].

In each site, soil samples were collected from 3 pits at a depth of 10 to 20 cm. We chose to sample at this subsurface depth because it is less sensitive than the surface to external (e.g., aeolian) input, and the microbial communities are not directly exposed to the extreme UV stress. Therefore, the shallow subsurface promotes preservation of biosignatures [54,55,56,57,58,59] and acts as an ideal sampling microhabitat. Samples for geochemical analyses were taken from each of the three pits with non-sterilized spades, homogenized and collected in clean plastic bags. Samples for cell amendment experiments were collected with ethanol-sprayed spades at the third sampling pit at each site, and stored in sterile Whirl-Pak® bags (Nasco, Fort Atkinson, WI, USA).

Rainfall, as an indicator of aridity in this extremely dry area, was the most important environmental variable we chose for site characterization. Annual precipitation information for each sampling site (Table 1) was determined from the nearest rain gauge(s) between the years of 1951 and 2017 (the sampling date) from Explorador Climático, Center for Climate and Resilience Research (explorador.cr2.cl/). We use this data as roughly representative of annual precipitation, but we acknowledge that gauging station data are sparse in the hyperarid core of the desert, and there may be local heterogeneity in rainfall distribution. The rain gauge reference(s) for site MES is Coya Sur (22.3939° S, 69.6225° W, 1250 m); for PONR-2 is Sierra Gorda (22.8892° S, 69.3219° W, 1616 m); for Yungay is Baquedano (23.3336° S, 69.8397° W, 1032 m); for TZ-0 are Aguas Verdes (25.4000° S, 69.9633° W, 1560 m) and Tal-Tal (25.4047° S, 70.4819° W, 9 m); for TZ-4 are Copiapo (27.3772° S, 70.3308° W, 385 m), Desierto de Atacama Caldera Ap. (27.2639° S, 70.7742° W, 204 m), and Caldera (27.0692° S, 70.8156° W, 15 m); for TZ-5 are Copiapo (27.3772° S, 70.3308° W, 385 m) and Elibor Campamento (27.7167° S, 70.1953° W, 750 m); and for TZ-6 are Elibor Campamento (27.7167° S, 70.1953° W, 750 m), Los Loros (27.8317° S, 70.1119° W, 940 m), El Totoral (27.9022° S, 70.9575° W, 150 m), Canto de Agua (28.0992° S, 70.7811° W, 330 m), Freirina (28.5072° S, 71.0806° W, 100 m), and Rio Huasco en Algodones (28.7306° S, 70.5067° W, 750 m). Daily rainfall at exact sampling location during the heavy rainfall event on the 6 to 7 June 2017 was computed by the Ventusky web application (www.ventusky.com/) using an Icosahedral Nonhydrostatic (ICON) Model, from the German Weather Service [60]. At the center of this heavy rainfall event, we predict 50–60 mm of rainfall, which we observed 6 months later as highly evaporated brines in topographic lows (Figure 2d).

### 2.2. Soil Characterization

Approximately 1 g of soil from each sampling pit was suspended in the 10× volume of 18.2 MΩ/cm resistance ultrapure water (Merck Millipore, Burlington, Massachusetts, USA). After agitating for 30 min at 300 rpm and 50 ° C, the pH was measured in triplicate by FiveGo pH meter F2 (Mettler Toledo, Columbus, Ohio, USA) and the electrical conductivity was measured in triplicate by HI-98129 Pocket EC/TDS meter (Hanna Instruments Ltd, Leighton Buzzard, England).

X-ray fluorescence (XRF) was conducted for major element determination. For XRF analysis, soil samples from each pit were crushed using a Planetary Micro Mill PULVERISETTE (FRITSCH, Germany) and sieved through a 355 μm sieve. The sieved samples were first combusted to eliminate all organics, sulfur, water and other volatiles. Ashed samples were fused with Lithiumtetraborate 50%/Lithiummetaborate 50% (T50/M50) (FLUXANA, Scientific and Medical Products Ltd, Stockport, England). Major elements were determined on fused pellets using a SPECTRO XEPOS XRF Spectrometer (SPECTRO Analytical Instruments, Kleve, Germany). Counting errors were determined in comparison with the errors in the calibration. The standard deviation of XRF analyses was between 0.01% and 0.05% for all elements, based on the determinations of BR, GSP-2, and NIST SRM 2711 standards [61].

Splits of soil samples were rinsed to remove dust and photographed at high-resolution under a VHX-2000 Digital Microscope (KEYENCE UK and Ireland) (Appendix A). Four view fields were randomly chosen from 10 g of each soil sample. The longest and shortest edges of 25 particles were measured from each view field using Fiji [62]. In total, 200 lengths of particles from each site were measured to estimate median grain size (Appendix A).

### 2.3. Nitrate and Ammonium Measurements

Soil samples for nitrate and ammonium concentration measurements were sieved with a 1.4 mm stainless steel sieve to remove the bulk of the silicate component. To extract the nitrate and soluble ammonium, sieved samples were suspended and mixed well in 18.2 MΩ/cm resistance ultrapure water using a 10:1 dilution factor, and shaken for 45 minutes at 200 rpm and 40 ° C. After centrifugation at 4000 rpm for 10 min, the supernatant of the soil suspension was filtered through a 0.20-micron filter. All implements that touched the samples were triple-rinsed in ultrapure water prior to contact. Nitrate concentrations were measured in triplicate on this filtered solution by Ion Chromatography (IC) via a Metrohm 930/889 IC + Autosampler (Metrohm AG, Switzerland), with a 150 mm Metrosep A Supp 5 separation column (4 mm bore), using 3.2 mM Na_2_CO_3/_1.0 mM NaHCO_3_ eluent at a flow rate of 0.7 mL/min. Soluble ammonium concentrations were also measured in triplicate by IC, with a 250 mm Metrosep C 4 separation column (4 mm bore) using 1.7 mM nitric acid_/_0.7 mM dipicolinic acid eluent at a flow rate of 0.9 mL/min. Peak areas were calibrated for nitrate and ammonium concentrations with 9 anion standards and 7 cation standards prepared in the laboratory and measured in triplicate by the same IC methods.

### 2.4. Total Organic Carbon (TOC) and Total Nitrogen (TN)

For TOC and TN analyses, soil samples were ground to a fine powder and decarbonated, by adding 2 M HCl and sonicating for 30 min. Extra HCl was removed by washing three times with ultrapure water. All soluble salts, including nitrate, were similarly washed away after decarbonation. Hence, our TN measurements do not include nitrate but only organic- and clay-bound nitrogen. TOC and TN were measured in an elemental analyzer (EA) Isolink coupled to a MAT253 isotope ratio mass spectrometer (IRMS) via a Conflo IV (Thermo Fischer Scientific, Bremen, Germany) in triplicate. Peak areas were calibrated for C and N abundances with multiple aliquots of the international reference material USGS-41.

### 2.5. Nitrate Amendments and Colony-Forming Units (CFUs)

About 10 g of sterilely collected soil from the third pit of each sampling site was amended with 4.5 mL of sterilized 10% sodium nitrate (~73,000 ppm NO_3_) and 4.5 mL of sterilized ultrapure water (as a control) and left for 4 days at 21 ° C. Tests showed that 4 days allowed for visible changes in the cultivable microbial colony number compared to in the control (without any amendments). Soils were stored at 4 ° C prior to culturing experiments to maintain the overall structure of the soil microbial community. We acknowledge various nutrient effects from different cell culture plates on the growth of cultivable aerobic heterotrophic microorganisms (which are only able to use exogenous organic sources and grow on laboratory-made culture plates), so three types of agar plates were selected in this experiment: (1) ultrapure agarose (15 g/L Agarose), with agarose as the only nutrient source; (2) tryptic soy agar (15 g/L Agar, 15 g/L Peptone, 5 g/L Soyabean Digest, 0.7 g/L Lecithin, 5 g/L Tween 80), with casein and plant-derived organics and glycerophospholipids as additional nutrients; (3) Luria–Bertani (LB) agar (15 g/L Agar, 10 g/L Tryptone, 5 g/L Yeast Extract, 5 g/L NaCl), with casein and yeast-derived organics and table salt as additional nutrients; and (4) plate count agar (9 g/L Agar, 5 g/L Tryptone, 2.5 g/L Yeast Extract, 1 g/L Dextrose,), with casein and yeast-derived organics and glucose as additional nutrients.

Duplicate amended soils and original soils without amendments were suspended in sterilized ultrapure water by an applicable dilution factor and spread on these three types of culture plates, sealed, and left at 21 ° C. Colonies were counted after 20 days of growth [63]. CFUs were calculated by multiplying the number of colonies formed on agar plates by the corresponding dilution factor, and a factor of 1.45 to account for the addition of 4.5 mL of solution to 10 g of soil. The relative microbial growth rate is defined as the difference of CFUs between samples inoculated with nitrate amendments and those treated with water only (controls) on a logarithmic scale to eliminate the effects caused by water input alone.

### 2.6. Hierarchical Clustering, Correlation and Linear Regression Analyses

Data from sampling sites were analyzed using the Past 3.26 software [64,65] by hierarchical clustering analysis using XRF results (Appendix A), TOC, TN, pH, electrical conductivity, grain size, and mean annual precipitation as primary environmental variables. Algorithm and distance matrixes were set to an unweighted pair group method with arithmetic mean (UPGMA) and Bray–Curtis dissimilarity index, respectively.

Bivariate correlation and linear regressive analyses were run using IBM SPSS Statistics 25 software [66] to examine statistical bases for the following: (1) how mean annual precipitation, grain size, and SiO_2_/Al_2_O_3_ ratios affect nitrate and TOC distribution; (2) how NO_3_^-^/TOC ratios affect the TN contents; and (3) how annual precipitation and nitrate affect total CFUs and the relative microbial growth rates with nitrate amendments. These variables were examined on a logarithmic scale to eliminate the bias skewed by the order of magnitude. Logarithmic variables in culturing experiments were examined with reliability analysis for internal consistency assessment. For bivariate correlation analysis, if the normal distribution tests (Kolmogorov–Smirnov and Shapiro–Wilk normality tests) of both samples were significant, the Pearson *r* correlation was used; if at least one sample was not normally distributed, the Spearman *ρ* correlation was used.

## 3. Results

### 3.1. Physical and Chemical Properties of the Soils

Bulk physical and chemical properties of the Atacama soils at 10–20 cm depth are listed in Table 1 and Table 2. The measured pH in our study sites was between 8 and 10, and electrical conductivity varied from 0.8 to 27 mS/cm. Median grain size varied from 291 μm in TZ-4 to 493 μm in TZ-0. All sites have a very positive skewness in grain size, with the lowest skewness in TZ-4 and the highest in Yungay (Appendix A).

On average, the sampled soils were composed of about 54.0% silicon (Si), 13.2% aluminium (Al), 7.9% calcium (Ca), 5.4% iron (Fe), 3.1% sodium (Na), 2.3% magnesium (Mg), 2.3% potassium (K), 0.65% titanium (Ti), 0.22% phosphorus (P), 0.11% manganese (Mn), and 0.01% chlorine (Cl) (Appendix A). The SiO_2_/Al_2_O_3_ ratios varied from 3.5 (TZ-4) to 4.5 (MES and TZ-5) (Table 2).

Nitrate concentrations were highly variable, ranging from 1 ppm (at TZ-0) to 7000 ppm (at TZ-4) (Table 2). Nitrate distribution varied amongst the individual sites, but did not show any significant correlation with either mean annual precipitation or the recent heavy rainfall event (Figure 3). Site TZ-4 in particular has an extremely high nitrate concentration. Soluble ammonium was below the limit of detection in all soils (Table 2). TOC and TN concentrations ranged from 40 to 1600 ppm and from 10 to 200 ppm, respectively (Table 2).

### 3.2. Cell Counts on Agar Plates

Results of CFU counts are shown in Table 3. On ultrapure agarose plates, CFU counts ranged from 0 (MES) to 1.62 × 10^3^ (TZ-4) without any amendments, from 0 (MES) to 3.57 × 10^3^ (TZ-4) on water control, and from 2 (MES and PONR-2) to 3.15 × 10^3^ (TZ-4) with nitrate amendments. On tryptic soy agar plates, CFUs ranged from 37 (MES) to 3.21 × 10^5^ (TZ-6) without any amendments, from 312 (TZ-0) to 1.65 × 10^6^ (TZ-4) on water control, and from 12 (MES) to 4.38 × 10^5^ (TZ-6) with nitrate amendments. On LB agar plates, CFUs ranged from 17 (MES) to 2.39 × 10^5^ (TZ-4) without amendments, from 7 (MES) to 1.23 × 10^6^ (TZ-4) on water control, and from 0 (MES) to 3.27 × 10^5^ (TZ-6) with nitrate amendments. Growth occurred on all plate count agar plates, with CFUs ranging from 167 (MES) to 2.90 × 10^6^ (TZ-4) without amendments, from 638 (TZ-0) to 1.33 × 10^7^ (TZ-4) on water control, and from 15 (MES) to 2.47 × 10^6^ (TZ-4) with nitrate amendments.

On ultrapure agarose plates, the cultivable aerobic heterotrophic microbial populations from the Yungay site decreased more than 10-fold when amended with nitrate solution compared to water-only amendments, while those from TZ-0 and TZ-5 increased more than 10-fold; PONR-2 cultivable soil microbiome slightly decreased with nitrate amendment, but MES and TZ-6 slightly increased. On tryptic agar plates, cultivable microbiomes from MES and TZ-4 decreased by one to three orders of magnitude with nitrate amendments; TZ-0 slightly decreased, but PONR-2, Yungay, and TZ-5 slightly increased. On LB agar plates, the cultivable microbial populations from Yungay and TZ-4 decreased one to three orders of magnitude with nitrate amendments; MES and TZ-0 slightly decreased, but TZ-5 and TZ-6 slightly increased. On plate count agar plates, cultivable microbial populations of MES, PONR-2, and Yungay decreased sharply 100- to 10,000-fold; all other sites decreased slightly (Table 4).

## 4. Discussion

### 4.1. Site Categorization

All of our sampling sites were located on or proximal to Mesozoic–Cenozoic sedimentary rocks and volcanics or Paleozoic–Cenozoic intrusives (Figure 1). The resulting pedological composition reflects the long-term hyperaridity of Atacama soils [21], and thus this study should shed light on the properties of indigenous extremophiles evolved over millions of years.

Based on essential soil physical and chemical properties, the intrasite heterogeneity of the seven study sites is less than 25% (Figure 4). Three clusters of sites emerge from the hierarchical clustering analysis, with more than 75% similarity (0.75 score) within each cluster: (1) sites with low annual precipitation (sites MES, PONR-2, Yungay, and TZ-0); (2) sites with relatively high annual precipitation (sites TZ-5 and TZ-6); and, (3) the northern-most transitional site, TZ-4, which exhibits some of the characteristics of each of the other clusters (Figure 4). We, therefore, refer to MES, PONR-2, Yungay, TZ-0 as “hyperarid” sites, TZ-5 and TZ-6 as “arid” sites, and TZ-4 as the “middle” site, which broadly captures the changes evident within other aridity indexes such as the evaporation/precipitation ratios and the percent cover of vegetation.

### 4.2. Nitrate Distribution and Effects on Microbial N Assimilation

Following the characterization above, the hyperarid sites of the Atacama Desert (MES, PONR-2, Yungay and TZ-0) can be described as an analog to the late Hesperian/Amazonian Periods [67] on Mars [18], with low moisture and large nitrate reservoirs. The transitional and arid sites, on the other hand (sites TZ-4, TZ-5 and TZ-6), are more analogous to regions of Mars where precipitation and organic C levels might have been slightly elevated, like Noachian Mars [67]. Therefore, comparisons between the hyperarid sites and transitional/arid sites provide insight into the strategies that potentially extant microorganisms on Mars might have uses to respond to changes in rainwater and nitrogen input.

Nitrate concentrations in our samples do not directly scale with precipitation estimates, as might be expected (Figure 3). Soils in our 10–20 cm sampling depth could have been influenced by the Yungay-region-centered unexpected heavy rainfall ~6 months before our sampling [36] (Figure 2d), since 2 mm of rainfall has been shown to influence the moisture content down to 5 cm depth in this region [68]. Measurements and modeling of highly soluble salts in the Mojave Desert soils [69] indicate that nitrate is transported to 1.3–2.7 m depth in profile, even in their driest site which experiences 84 mm/yr of rain, a distribution which rarely involves a single rainfall event in excess of 50–60 mm. Therefore, the massive rainfall events could presumably have leached away the majority of highly soluble nitrate in the shallow subsurface of the affected sites, contributing to the high inter-site variability in the distribution of nitrate (Figure 3b). It is certainly possible that the massive rainfalls did not affect the middle site TZ-4 as significantly, although dead brush and the presence of small desiccated roots hint that a desert bloom may have occurred at the site prior to our sampling, which could also have affected near-surface nitrate levels. In addition, ammonia-oxidizing bacteria are active during desert blooms, which could also have affected nitrate concentrations [70].

However, additional physicochemical factors, such as sedimentation in a high-energy environment with high wind speed [30], could also account for this seemingly random nitrate distribution (Figure 3). The middle site TZ-4 has ten to one hundred times higher nitrate concentrations than the other sampled Atacama sites, but it also has smaller average grain sizes and lower SiO_2_/Al_2_O_3_ ratios (Table 2). Sedimentation rates, which in this case, mostly reflect the influx of wind-blown dust, as indicated by the moderate roundedness and frosting of the grains (Appendix A), can be inferred from grain size distribution and SiO_2_/Al_2_O_3_ (approximating quartz to clay [71]) ratios in these soils [72,73]. This proxy is based on the assumption that quartz is heavier (~2.65 g/cm^3^ compared to clay density ~1.33 g/cm^3^) and thus less likely to be deposited in a low-energy environment where only small particles are transportable. This comparison, therefore, suggests that site TZ-4 has experienced lower sedimentation rates than other sites sampled. Lower sedimentation rates would mean correspondingly less dilution of atmospheric nitrate by terrigenous input, and correlatively higher nitrate levels. Bivariate correlation analyses suggest that grain sizes (Appendix A and Table 1) (*ρ* = −0.42 *, *p* = 0.028) and SiO_2_/Al_2_O_3_
*(ρ* = −0.37 *, *p* = 0.047) are more directly correlated with nitrate concentrations at all sampling sites compared to annual precipitation (*ρ* = −0.19, *p* = 0.206). Moreover, the nitrate that is photolytically synthesized from gaseous nitric oxides is found adsorbed by aluminium-bearing [71] aerosols and dust grains [27,74]. Therefore, we speculate that in addition to annual precipitation, other physical factors such as sedimentation rates can also significantly influence the nitrate distribution in Atacama, and, by proxy, the more recent dry Martian soils.

Atacama soils are generally assumed to be C-limited with very slow carbon cycling [75], because of low organic C/N ratios. Phylogenetically ancient eukaryotes related to red algae [76,77], microbial biofilms [78], and specifically UV-resistant microorganisms [79,80] are also found to be present in the Atacama Desert, and could provide a persistent source of organic carbon for heterotrophy [81]. However, microbial C mineralization remains at low levels even with episodically enhanced moisture input associated with rainfall events [55,82]. Compared to less arid regions, the microbial communities from the hyperarid core usually have higher exogenous organic C uptake activities [82]. With little external input, leaching and subsurface decomposition [75], organic C in Atacama soils should mostly be derived from in situ carbon fixation.

The distribution of TOC increases along with annual precipitation (Figure 5a), suggesting that water is a limiting factor for microbial growth, as expected. Total nitrogen (TN) is positively correlated with TOC (Figure 5b), indicating that the majority of nitrogen in these soils is organic-bound rather than ammonium, which could substitute for potassium in detrital clay minerals formed during diagenesis [83]. The intercept of this plot at 40 ppm could indicate a small fraction of clay-bound N derived from in situ degradation of biomass, analogous to processes described from marine sediments [84,85]. This interpretation is validated by a lack of correlation between TN and potassium (Appendix A), as well as the low abundance of ammonium pool (Table 2) [35]. We can, therefore, refer to the non-soluble N in our soils as total organic nitrogen (TON).

Furthermore, an anti-correlation between SiO_2_/Al_2_O_3_ ratios and TOC in the hyperarid core of the Atacama Desert (with ≤2 mm/yr rainfall; Figure 5c) suggests that lower quartz/clay ratios lead to less dilution of the soil TOC contents produced by microbial communities in situ. Given the potential dilution effect of sedimentation rates on dust-bound nitrate concentrations discussed above, these results suggest that a greater proportion of clays (i.e., lower quartz/clay ratios) in hyperarid sediments could also enhance biomass preservation, consistent with previous studies suggesting that clay minerals could constitute a microniche suitable for the preservation of biochemical materials [86,87,88].

The TOC content in these soils can be concomitantly enriched and redistributed by increasing rainfall. However, in general, TOC acts as an approximate quantification of the amount of biomass actively affected by ambient nitrate, even in such small proportions [58]. In order to access nitrate, microorganisms generally need water to dissolve and solubilize nitrate salts and other bioessential molecules in the soil. In more humid environments, nitrate availability to organisms generally scales with nitrate richness, leading to higher microbial abundance at higher nitrate levels. At the hyperarid sites of the Atacama Desert, where rainfall is almost absent, multiple regression analyses suggest that TOC (approximating biomass) is positively associated with both annual precipitation (standardized coefficient = 1.02***, *p* < 0.001) and nitrate concentrations (standardized coefficient = 0.53*, *p* = 0.014). These trends suggest that nitrate concentrations remain as important as rainfall for microbial abundance in hyperarid Mars-analogous environments.

On the other hand, when annual precipitation exceeds 10 mm, the positive effect of nitrate on microbial abundance declines (Figure 5d). In spite of the strong rainfall control on microbial metabolic activities, nitrate to TOC ratios (as a representative of the availability of nitrate to microbes) are positively correlated with TON in the hyperarid (*r* = 0.60*, *p* = 0.019) and arid sites (*r* = 0.82*, *p* = 0.023) we sampled. This suggests that microbial communities with a higher probability to contact nitrate can assimilate nitrate more quickly for metabolism and biomass production, regardless of water availability.

In summary, our results suggest that both the mean annual precipitation (as a long-term regulator of both nitrate and biomass) and sedimentation rates can influence nitrate concentrations in hyperarid soils. Sedimentation rates can additionally dilute TOC. The significant positive correlation between the indigenous nitrate concentrations and TON contents indicates that this nitrate is likely bioavailable and assimilated by microorganisms.

### 4.3. Growth of Cell Cultures with Nitrate Amendments 

We tested the importance of nitrate to desert extremophiles using laboratory growth experiments constrained by CFUs inoculated from Atacama soils. MES generally had the lowest CFUs, while TZ-4 had the highest CFUs (Table 4). CFUs of TZ-6 exceed TZ-4 on tryptic soy agar and LB agar plates with nitrate amendments. TZ-0 CFUs become lower than MES on tryptic soy agar and plate count agar plates with water amendments only. These findings qualitatively implicate that (1) MES has the least active microbiome, and might constitute the driest site studied [16]; (2) TZ-4 has the most abundant active microbiome, probably as a consequence of its highest nitrate availability; (3) the cultivable microbial population at site TZ-6 benefits the most from excessive nitrate addition; and (4) the cultivable extremophiles at TZ-0 respond neutrally to moisture elevation (Table 4), indicating a small but stable microbial community.

After logarithmic transformation, the Cronbach’s *α* of reliability analysis is 0.85 among annual precipitation, nitrate, CFUs, and the microbial response pattern to nitrate amendments, indicating an acceptable internal consistency of samples. By nonparametric bivariate correlation, positive correlations are determined between annual precipitation and total cultivable CFUs (ultrapure agarose plate *r* = 0.88**, *p* = 0.004; tryptic agar plate *r* = 0.93**, *p* = 0.001; LB agar plate *r* = 0.83*, *p* = 0.010; plate count agar plate *r* = 0.69*, *p* = 0.042). By multiple regression along with mean annual precipitation as a confounding variable, local nitrate concentrations also positively exert influence on CFUs on tryptic soy agar (standardized coefficient = 0.30*, *p* = 0.050), LB agar (standardized coefficient = 0.54**, *p* = 0.001), and plate count agar plates (standardized coefficient = 0.60*, *p* = 0.041), but not on ultrapure agarose plates. However, the distribution of cultivable aerobic heterotrophic CFUs in general illustrates that no microbial communities show a significant preference for excessive nitrate amendments other than TZ-0 and TZ-5 on ultrapure agarose plates, if more than one order of magnitude change is defined to be significant. These trends might suggest that the growth of cultivable microbes on ultrapure agarose, as the least nutrient culture medium, is more easily affected by external factors.

In comparison to the arid sites and TZ-0, nitrate amendments have a more negative effect on the relative microbial growth rate at the hyperarid sites and TZ-4, with the exception of tryptic soy agar plates (Table 4), probably because of the higher proportion of nitrogenous compounds in soyabean digest than in yeast extract [89,90]. These results, to some degree, seem to confirm that in the wetter sites with relatively low nitrate levels (e.g., TZ-5 and TZ-6), native microbial communities generally have a higher nitrate tolerance and sometimes prefer higher nitrate inputs; while in drier sites, extreme excess nitrate may inhibit denitrification [91] and sulfate reduction [92], and further inhibit growth [93] by causing damage to microorganisms due to the osmotic stress [94]. Thus, this association between annual precipitation and the relative microbial growth rates is presumably a result of nitrate bioavailability enhanced by solubilization during precipitation. This is also supported directly by biologically altered stable oxygen isotopes of nitrate [35], and indirectly by the higher microbial carbon mineralization activity in less arid regions [82].

Previous work has identified a wide range of microorganisms persisting in Atacama Desert soils, even within hyperarid areas. The growth and enriched nitrate response patterns of microbial colonies in this culturing experiment might reflect viable cultivable species from these previously identified microorganisms. Among them, *Actinobacteria* are generally the predominant bacterial phylum in the hyperarid core of the Atacama Desert [54,56,63,95,96], with most of the detected species belonging to the genus *Frankia* [95]. Other common dominant bacteria are from *Chloroflexi* [97], *Gemmatimonadetes*, *Planctomycetes* [98], *Proteobacteria*, *Firmicutes* [58], *Aquificae*, and *Deinococcus-Thermus* [54]. *Cyanobacteria*, especially those from genus *Chroococcidiopsis*, have been identified in both hypolithic [99,100,101] and endolithic environments [102,103]. Prokaryotic species found in the surface layer of the Atacama Desert include more irradiation-tolerant bacteria, such as *Geodermatophilaceae* and *Rubrobacter*, while more halophilic or halotolerant bacteria (e.g., *Comamonadaceae*, *Bacillaceae*, and *Alicyclobacillaceae*) and archaea (e.g., *Halobacteria*) were usually detected at depths of 20 to 100 cm [55]. Many of these taxa can perform denitrification [28,104], assimilatory and dissimilatory nitrate reduction [105,106,107] (Kyoto Encyclopedia of Genes and Genomes, [108]). These microorganisms are the main contributors of organic carbon, and potentially the beneficiaries of surrounding nitrate sources.

## 5. Conclusions

Here, we use a combination of geochemical and microbiological techniques to investigate the effects of atmospherically derived nitrate on biomass and microbial growth in an infertile Mars analog natural setting (the Atacama Desert) with extremely low precipitation, high UV irradiation, and large nitrate reservoirs. It is intriguing to discover that higher nitrate to biomass ratios seem more beneficial to microbial growth and N assimilation than precipitation within all sites sampled. Both precipitation and dilution of nitrate with terrigenous sediments seem to be important controls on nitrate concentrations in arid regions. On Mars, nitrate should be primarily regulated by sedimentation rates and act as the only N source to microorganisms (probably irradiation-resistant and halophilic prokaryotes [55]). Our results from Atacama soils suggest that potentially extant Martian life could survive under the high levels of nitrate availability [3,33,34]. On Mars, the concentrations of nitrate and TOC should be even more tightly regulated by local sedimentation rates after the early Martian periods dominated by impact thermal shocks [40,41,42], since the more recent dry Mars does not have detectable rainfall. With the equilibrium between Martian nitrate and atmospheric nitrogen stabilized by impact thermal shocks and without rainwater to leach nitrate away, the nitrate availability near the surface on the more recent dry Mars remains nearly unchanged. Hence, any organisms present in the shallow subsurface of Mars should not be experiencing the high nitrate stress suggested by our culturing experiments. We suggest that sustained high nitrate availability in Martian sediments could potentially provide a sufficient nutrient source to support life over geological timescales spanning the decline of the wet early Mars to the more recent hyperarid Mars. In addition, high nitrate levels could be a sign of habitability, providing a useful guide for future life detection on Mars.

## Figures and Tables

**Figure 1 life-09-00079-f001:**
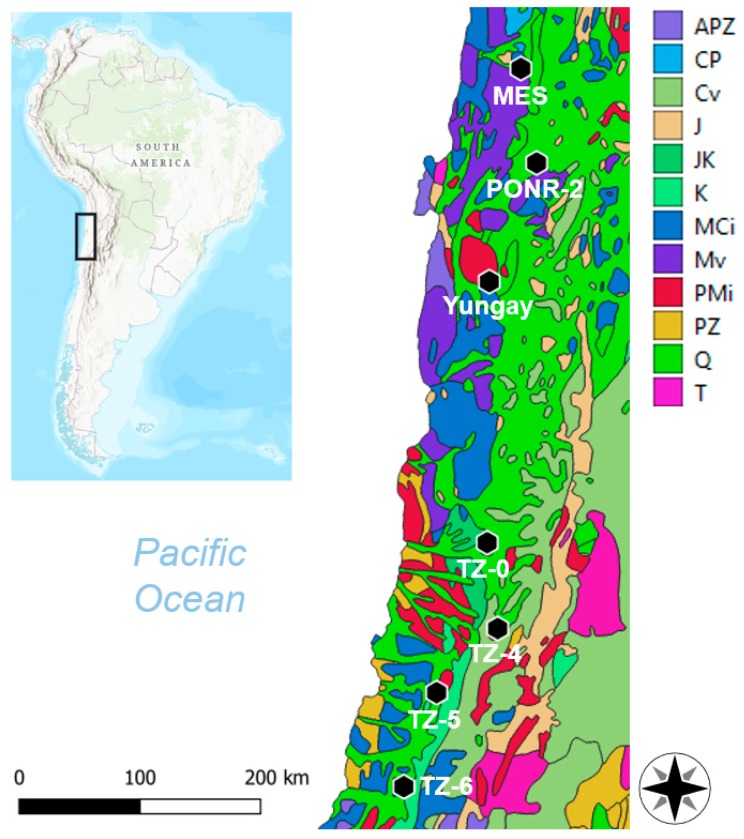
Seven study sites (María Elena South [MES], PONR-2, Yungay, TZ-0, TZ-4, TZ-5, and TZ-6) from the Atacama Desert, Northern Chile. APZ, Precambrian–Paleozoic sedimentary rocks; CP, Carboniferous–Permian sedimentary rocks; Cv, Cretaceous–Tertiary volcanics; J, Jurassic sedimentary rocks; JK, Jurassic–Cretaceous sedimentary rocks; K, Cretaceous sedimentary rocks; MCi, Mesozoic–Cenozoic intrusives; Mv, Mesozoic volcanics; PMi, Paleozoic–Mesozoic intrusives; PZ, Paleozoic sedimentary rocks; Q, Quaternary sedimentary rocks; T, Tertiary sedimentary rocks [52].

**Figure 2 life-09-00079-f002:**
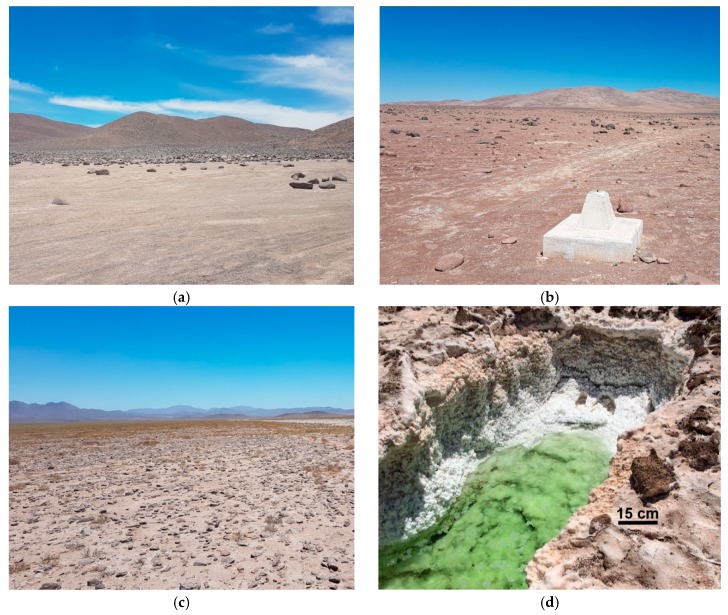
Landscape views of (**a**) sampling site in María Elena South (MES), (**b**) sampling site in Yungay, (**c**) sampling site in Transition Zone 4 (TZ-4), and (**d**) brackish pool in Yungay region formed after 50–60 mm rainfall (illustrating a rough penetration depth and scale bar).

**Figure 3 life-09-00079-f003:**
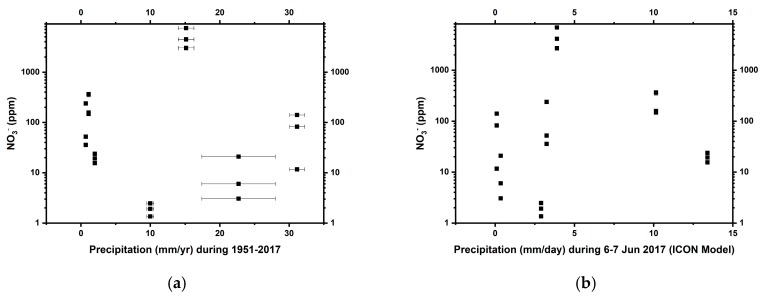
The distribution pattern of nitrate as a function of (**a**) mean annual precipitation (1951–2017) and (**b**) ICON-modeled daily precipitation (6–7 Jun 2017) in the Atacama Desert. Errors on nitrate (NO_3_^-^) measurements were smaller than the symbols.

**Figure 4 life-09-00079-f004:**
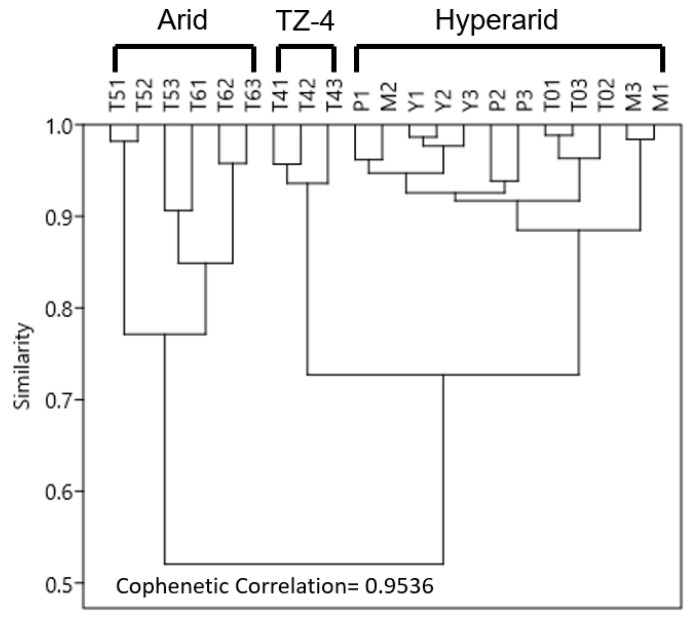
Hierarchical clustering (UPGMA algorithm, Bray–Curtis distance matrix), which shows intra- and inter-site heterogeneity of seven study sites based on major elements in the regolith, total organic carbon (TOC), total nitrogen (TN), pH, electrical conductivity, grain size, and annual precipitation.

**Figure 5 life-09-00079-f005:**
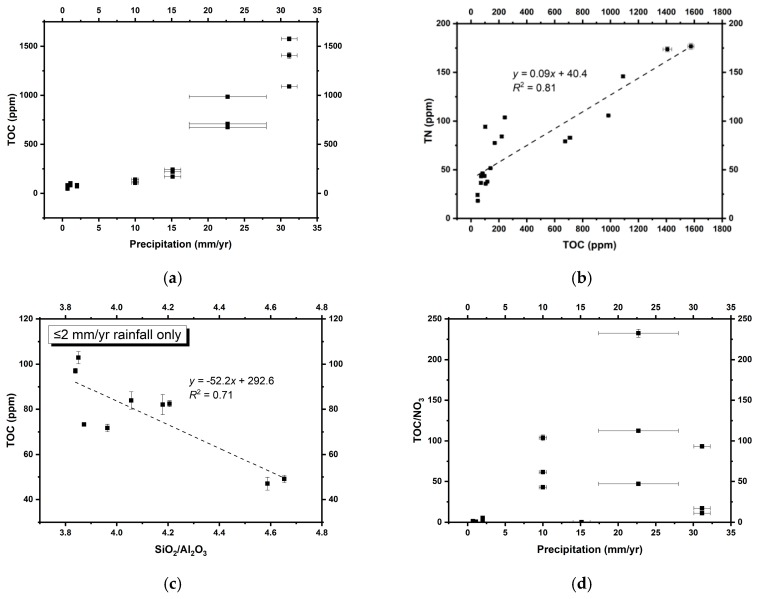
(**a**) TOC versus inferred annual precipitation; (**b**) TOC versus TN; (**c**) TOC versus SiO_2_/Al_2_O_3_ (approximating quartz/clay) ratios in the core Atacama Desert; (**d**) annual precipitation versus TOC/nitrate ratios. Errors on some TOC, TN, and SiO_2_/Al_2_O_3_ measurements were smaller than the symbols.

**Table 1 life-09-00079-t001:** Site characterization, showing terrestrial coordinates, altitude, estimated annual precipitation, modeled daily precipitation during the heavy rainfall event in June 2017, and median grain sizes.

Site	Lati-Tude	Longi-Tude	Alti-Tude (m)	Annual Precipi-Tation (mm/yr)	Modeled Precipitation (mm/day) (6–7 June 2017)	Median Grain Size (μm)
MES	22.2641° S	69.7243° W	1493 ± 8	0.7	3.3	444
PONR-2	23.0726° S	69.5892° W	1493 ± 8	1.1	10.1	486
Yungay	24.0884° S	69.9945° W	1007 ± 10	2.0	13.4	464
TZ-0	26.3222° S	70.0128° W	1106 ± 10	10.0 ± 0.5	2.9	493
TZ-4	27.0565° S	69.9228° W	1658 ± 10	15.1 ± 1.1	3.9	291
TZ-5	27.6051° S	70.4458° W	588 ± 8	22.7 ± 5.3	0.4	322
TZ-6	28.4100° S	70.7270° W	658 ± 6	31.2 ± 1.1	0.1	416

**Table 2 life-09-00079-t002:** Characterization of sampling pits, showing means ± standard errors, if applicable, of pH, electrical conductivity, concentrations of nitrate, total organic carbon (TOC), and total nitrogen (TN), and the silicon to aluminium content. We refer to the three sampling pits from MES as M1, M2, M3; PONR-2 as P1, P2, P3; Yungay as Y1, Y2, Y3; TZ-0 as T01, T02, T03; TZ-4 as T41, T42, T43; TZ-5 as T51, T52, T53; TZ-6 as T61, T62, T63.

Pit Name	pH	Conductivity (mS/cm)	NO_3_^−^ (ppm)	NH_4_^+^ (ppm)	TOC (ppm)	TN (ppm)	SiO_2_/Al_2_O_3_
M1	8.96 ± 0.00	1.41 ± 0.07	35.7 ± 0.3	<0.5	49.1 ± 1.7	18.0 ± 0.4	4.65
M2	8.29 ± 0.02	8.69 ± 0.08	238.4 ± 3.6	<0.5	82.0 ± 4.5	45.2 ± 1.1	4.18
M3	8.38 ± 0.01	9.69 ± 0.09	52.1 ± 0.2	<0.5	47.0 ± 2.8	24.1 ± 1.9	4.59
P1	8.29 ± 0.01	12.97 ± 0.03	147.5 ± 2.5	<0.5	82.5 ± 1.3	46.2 ± 0.2	4.21
P2	8.20 ± 0.01	22.68 ± 0.11	362.0 ± 39.1	<0.5	102.9 ± 2.7	94.2 ± 1.9	3.85
P3	8.09 ± 0.00	22.10 ± 0.12	157.7 ± 2.6	<0.5	97.1 ± 1.1	43.5 ± 0.9	3.84
Y1	8.03 ± 0.00	20.45 ± 0.20	23.8 ± 0.1	<0.5	73.3 ± 0.6	43.2 ± 0.1	3.87
Y2	8.09 ± 0.00	17.14 ± 0.07	19.3 ± 1.0	<0.5	71.7 ± 1.6	36.4 ± 1.4	3.96
Y3	8.11 ± 0.00	17.66 ± 0.16	15.5 ± 0.3	<0.5	83.9 ± 3.8	45.5 ± 2.1	4.06
T01	9.29 ± 0.01	0.86 ± 0.01	1.9 ± 0.0	<0.5	117.6 ± 2.1	37.7 ± 0.6	3.91
T02	9.37 ± 0.01	0.89 ± 0.06	1.4 ± 0.0	<0.5	140.6 ± 4.9	51.6 ± 0.4	4.13
T03	9.83 ± 0.00	1.06 ± 0.06	2.5 ± 0.0	<0.5	106.1 ± 4.2	35.6 ± 0.3	3.77
T41	8.39 ± 0.01	13.62 ± 0.38	2686.8 ± 141.6	<0.5	220.8 ± 7.4	84.1 ± 1.2	3.76
T42	8.07 ± 0.01	26.02 ± 0.48	6802.6 ± 290.2	<0.5	242.5 ± 2.0	103.7 ± 0.9	3.55
T43	8.26 ± 0.00	17.20 ± 0.28	4100.0 ± 160.4	<0.5	170.6 ± 2.3	77.4 ± 0.8	3.66
T51	9.60 ± 0.00	0.82 ± 0.06	3.0 ± 0.1	<0.5	708.7 ± 14.8	82.9 ± 1.8	4.49
T52	8.96 ± 0.01	4.08 ± 0.13	6.0 ± 0.1	<0.5	674.9 ± 7.2	79.0 ± 0.7	4.44
T53	8.79 ± 0.00	20.24 ± 0.45	20.9 ± 0.2	<0.5	985.3 ± 2.4	105.7 ± 0.5	4.47
T61	9.46 ± 0.00	1.69 ± 0.09	11.7 ± 0.2	<0.5	1089.9 ± 12.3	145.9 ± 1.3	4.23
T62	8.60 ± 0.01	16.23 ± 0.35	140.6 ± 1.3	<0.5	1576.1 ± 21.4	176.7 ± 3.0	4.36
T63	9.20 ± 0.00	6.07 ± 0.14s	82.4 ± 0.8	<0.5	1407.2 ± 30.7	173.6 ± 2.5	4.32

**Table 3 life-09-00079-t003:** Colony-forming units (CFUs) on ultrapure agarose, tryptic soy agar, Luria–Bertani (LB) agar, and plate count agar plates without amendments, amended with water only, and amended with 10% sodium nitrate.

Type of Culture Plate	Amendment	M3	P3	Y3	T03	T43	T53	T63
Ultrapure agarose	None	0	17	13	93	1.62 × 10^3^	80	1.42 × 10^3^
	Water	0	15	106	111	3.57 × 10^3^	87	667
	Nitrate	2	2	7	1.33 × 10^3^	3.15 × 10^3^	1.62 × 10^3^	2.93 × 10^3^
Tryptic soy agar	None	37	53	169	514	1.60 × 10^5^	1.33 × 10^5^	3.21 × 10^5^
	Water	5.76 × 10^3^	3.39 × 10^3^	1.14 × 10^4^	312	1.65 × 10^6^	5.22 × 10^4^	5.09 × 10^5^
	Nitrate	12	1.48 × 10^4^	5.20 × 10^4^	123	8.41 × 10^4^	7.61 × 10^4^	4.38 × 10^5^
LB agar	None	17	28	33	91	2.39 × 10^5^	2.95 × 10^3^	5.41 × 10^4^
	Water	7	7.98 × 10^3^	3.71 × 10^3^	491	1.23 × 10^6^	1.17 × 10^3^	1.54 × 10^5^
	Nitrate	0	9.43 × 10^3^	197	290	4.88 × 10^3^	1.00 × 10^4^	3.27 × 10^5^
Plate count agar	None	167	190	5.17 × 10^3^	353	2.90 × 10^6^	6.00 × 10^3^	2.72 × 10^5^
	Water	1.36 × 10^3^	8.56 × 10^4^	6.96 × 10^4^	638	1.33 × 10^7^	1.25 × 10^5^	4.12 × 10^6^
	Nitrate	15	44	29	343	2.47 × 10^6^	6.96 × 10^4^	5.22 × 10^5^

**Table 4 life-09-00079-t004:** Results of cell cultures on ultrapure agarose, tryptic soy agar, LB agar, and plate count agar plates with or without excessive nitrate amendments, illustrating the change in order of magnitude of CFUs with nitrate amendments (all variables are scaled by logarithmic transformation). Negative effects of nitrate on microbial growth more than one, two, and three orders of magnitude are labeled with *, **, and ***, respectively.

Pit Name	Ultrapure Agarose Plates	Tryptic Soy Agar Plates	LB Agar Plates	Plate Count Agar Plates
M3	0.38	−2.68 **	−0.86	−1.97 *
P3	−0.78	0.64	0.07	−3.29 ***
Y3	−1.17 *	0.66	−1.28 *	−3.38 ***
T03	1.08	−0.40	−0.23	−0.27
T43	−0.05	−1.29 *	−2.40 **	−0.73
T53	1.27	0.16	0.93	−0.25
T63	0.64	−0.07	0.33	−0.90

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
