# Peer review of "Nitrates as a Potential N Supply for Microbial Ecosystems in a Hyperarid Mars Analog System"

_life, 2019, doi:10.3390/life9040079_

Round 1
Reviewer 1 Report
Review report: “Atmospheric Nitrate as a Potential Nutrient for Life on Mars”
This paper aimed to identify whether nitrates were a limiting factor for microorganisms in Martian analogue sites across the Atacama Desert, Chile. It collected both observational data from the seven sample sites and experimental data by growing the microbial communities under different nitrate regimes. The authors overall concluded that nitrate levels on Mars are sufficient to host Earth-like life.
Whilst there is a lot of useful and unique data collected for this study, the analyses and written presentation in the paper need improvement. The following are areas or points to improve:
Title is misleading – nitrate is not tested as a potential nutrient and while there are applications to Mars the study was not focused on the Martian environment. Atmospheric nitrogen on Mars is lower than Mars but they tested limitation by addition of nitrates. Lines 59-75 – go into detail about phyla seen in the Atacama but this is not brought in at any point later. Overall aims of the study are very vague – consider splitting into two clear objectives: the observational study and the experimental study. Figure 1 is repeated but slightly different – I assume the first is an error. How far were the weather gauging stations from the sites – can we have a max and min distance? Line 111 – rainfall chosen for site characterisation, is this annual rainfall? And what is the justification for this given no detectable rainfall on Mars? Table one has no error on values, especially those measured in triplicate. It is standard practise to use the median to measure grain size instead of mean (especially looking at the skewed histogram in figure S1 this is necessary). The units used for nitrogen and carbon, ppm, are non-SI units. Finally the NO3 value for TZ-4 seems unlikely as it is not only high but a round 5000 – this reads like a measurement error so standard errors should be shown. Line 121-123 - soils characterisation methods are incomplete. pH and EC are measured by water suspension, which is not recorded here. Also the measurements in “triplicate” is not explained – how were these triplicates selected, are these averaged across the three pits sampled from and how were they averaged? For the particle measurement in lines 133-137 it is not made clear how these views were randomly selected, as with grain sizes where the view field was taken from a core can greatly impact the sizes seen due to the self-sorting properties of coarse soils. Also unclear is whether these samples were from one pit or a homogenised sample across the whole site. In section 2.5 it is not made clear (nor elsewhere in the discussion) that some of the microorganisms from the sample will not grow on culture, especially due to the extremity of the environment in the Atacama. Line 161 – what is the molarity of the 10% sodium nitrate addition and what is the justification for using this amount? Line 162 – what is the justification for refrigerating the sample soils after the 4 days. Has the nitrates already present in the soil been taken into account when suspending the soils in solutions – could it be that soluble nitrates already in the soil are being plated alongside the nitrate amendments, thus oversaturating the system? This should be made clear in whichever case. The addition of salts is attributed to changes in CFUs but could this be an artefact of enhanced recovery from the soil matrix due to salt disruption? Please address this. Line 180 – mentions of a humidity gradient and yet no humidity is measured – are you referring to precipitation, which is not the same, or something other? Section 2.6 PC2 is not described. Path analysis is perhaps not the best model for this study due to its fairly small sample size and the large number of assumptions needed. A general linear model may be better as this will also fit gradient data. If path analyses are still to be used there should be a larger sample size, fit statistics for both of the models and quality controls. The description in the methods section would be better served as a legend for figures 6 and 7 and the errors written as “other” is unorthodox and not well described – the description of this in lines 208-213 is incorrect. When describing Atacama soils, gypsum/bassanite content could be useful as an indicator of soil geochemistry and moisture content. Section 3.1 has no comparative statistics, which would be more useful than the descriptions give, which just reiterate table 1. If there is a significant result, or something important then this should be indicated to the reader. Authors mention no correlation between rainfall and nitrate distribution and yet continue to work on the assumption they are link in figures 6 and 7. Figure 2 would be better if the sites were averaged and error bars going along both axes. 2b shows the rainfall during the anomalous event but not during the time of collection – there is no reason given for this. Figure 3 – how are these clusters assigned? There are no statistics given so are they just arbitrarily assigned by eye, in which case they are not robust enough to draw conclusions from. Section 3.3, including table 2, does not show any statistics. A brief description of the important comparisons or significant changes highlighted in bold would be useful here. Lines 263-266 are a justification and would be better suited for section 2.1. While the justification makes a small amount of sense, there is no mention of how small precipitation events often don’t penetrate to these depths, which given the importance placed on precipitation may be a confounding element in this study. Line 275 mentions humidity in relation to Mars – as the study acknowledges the humidity but lack of precipitation on Mars then why not measure humidity as a variable? Desert blooms (line 281) should have been introduced in the introduction if it is a confounding event as the impact is not fully described in the discussion. Line 284 – it is inferred that TZ-4 is a “more humid” site but there is no measure of humidity, nor does it say what it is more humid than or how they inferred this. Discussion has many results brought up in it – these should be more clearly separated. Line 295 p=0.000 is not possible, this should be p<0.001. Line 300 approximates TOC as quantification of biomass but ancient TOC can also be detected especially due to the highly preservative nature of Atacama soils, so this is not a clear indication of active biomass. Figure 4 – equations need and fit statistics should be made available for all of the graphs. The line in 4d is not convincing enough to base the conclusions drawn in lines 326-328 upon. Figure 5 – there aren’t enough points for a correlation to be drawn as these are highly skewed by the site replicates, especially in 4b with site TZ-4 (as evidenced by the scale). CFU data is skewed across plates – the “negative effect” mentioned in line 355 is only based on plate count agar plates (but this has to be inferred by reading supplementary material and not explicitly mentioned). The statement on line 357-360 seem like a stretch as they are based on one point and draw on references 67-69 which are studies of macro- rather than micro-organisms. Section 4.3 could be longer if the previous allusions to Mars in the discussion were moved here instead – this would lend more coherency to the text. The periods of Martian natural history are not common knowledge and the descriptions given in section 4.3 would be best suited for the introduction. It is only at line 384 that the lack of rainfall on Mars is addressed – this should be mentioned much sooner. When this is addressed it should also mention there are other potential source of water on Mars (such as atmospheric humidity and potential below-ground water systems) – it is not necessary that the water needs to be “rainwater”. There is no mention of dry nitrate deposition in this paper – which is relevant both to Mars and the Atacama. Too much emphasis has been put on precipitation. Reference to figure 6 in the conclusions is unnecessary – not only should conclusions be able to stand alone from figures, with just a few take-home messages, but also that figure does nothing to support the previous statement. Figure S1 the histograms have no axes. The pictures show the grains to be irregular in shape with no explanation as to which length was measured – an example of measurements taken may be useful here (ie measurement lines imposed onto the image). Table S2 – it is unclear where the NO3 measurements have come from as they do not match those on table 1. Also this table is referenced in the text in the context of the sites, so an additional column including the site name would be useful. References include many with incorrect doi links – these should be corrected.
Reviewer 2 Report
This study is a very interesting one when it comes to the combination of analytical, experimental and statistical means. It is very rare to see a geobiochemical study that puts that much effort in statistical solutions like path analysis. Overall, the presentation of the data and reasoning behind it is sound. Sometimes the explanation for why things are done is not sufficient (even if the performed experiment is presumably a standard test in the field). Example: CFUs - what is the reasoning behind the different plate agarose media, what do the authors read from the different counts on agarose, LB or plate count agar. This study presents a good start to dig deeper into the dependencies of soil properties and cell count.
Apart from some details that will described in the following, I find this study interesting and sound enough to be published – hopefully providing an example for similar bridging and spatially inclusive studies.
Line 109: maybe split figure 1 in a and b
Table 1: as TOC and TN are only explained later in the text, putting the explanation in the legend saves the reader the jump.
Line 165: Lysogeny broth is to my knowledge the more common term for LB medium
Line 237: Just because PC 2 probably has a higher weight for all variables, it might be interesting to know how the PCA would look like for PC 1 and PC 2 (maybe SI, but not absolutely necessary).
Line 304: For an outsider, the sentence is a bit misleading/information is missing to make the connection: how would Nitrogen substitute for potassium?
Figure 4d: the trend curve doesn’t really follow the trend. Also, the figure legend could explain a little bit better what is exactly displayed.
Figure 6: Legend could have more explanation on the visuals: weight regression numbers and level of statistical significance (like in the rest of path analysis diagrams).
Figure 7: “PCA” is a confusing abbreviation for plate count agar with principal component analysis in same text.
Reviewer 3 Report
This paper performs statistical analysis of Atacama data (PCA) to look for relationships between variables such as precipitation, TOC, TN, quartz to clay ratios, etc. In addition, viability of cultured organisms in amended (water or nitrate) and unamended Atacama surface materials is assessed. The results are interpreted with relevance to Mars.
While there is some interesting data here, I feel more work is needed to understand the significance of the differences between the different Atacama sites and that additional considerations are necessary in the application of this data to Mars.
To the first point, I am not a statistician and am not sure whether the application of PCA to a small dataset is appropriate. More importantly, the paper lacked a discussion of the heterogeneity within single sites, which made it difficult to assess whether the data for each site was representative of that site or not. How significant are the differences between sites, compared to the variation at the same site? How many samples were taken at each site?
Some of the correlations shown are not particularly strong as evidenced by R2 and in some cases, such as Fig. 5a, the regression may be strongly affected by an outlier which, if removed, would result in a poorer correlation.
To the second point, which is the relevance of the data to Mars, the paper would benefit from more discussion of the potential sources of nitrate on Mars and its potential transport by aqueous processes on ancient Mars. The paper states that nitrate availability is largely controlled by sedimentation, but in situ data suggests that nitrate may be leached by post-depositional processes. Please see below for specific comments.
I suggest a fairly major revision prior to publication to address the above issues.
Specific comments:
Abstract: Nitrate is “rich” l. 9
32 Nitrate salts “form an abundant component”
This is all relative. Nitrate is not abundant compared to water, sulfate, or even carbon bearing species in Mars soils. I would hesitate to use these terms and simply state that it is present at ppm abundance.
34 Exoplanet studies? Atacama is a Mars analog, period. I am not familiar with the exoplanet studies and the reference Navarro Gonzalez isn’t about exoplanets.
52-53 “analogous processes are hypothesized to occur on Mars” – yes this is true, but nitrate formation on Mars is as equally likely to have all happened during late heavy bombardment by thermal shock from impacts (See Manning et al. 2008, 2009, Navarro Gonzalez et al. 2019). This possibility is not acknowledged in the paper – there is actually no evidence to confirm modern nitrate deposition on Mars, although it may be happening. The paper should acknowledge the impact generation hypothesis. This may not change the paper’s hypotheses and analyses much but it should be acknowledged. The main point is that the Atacama nitrate deposits may be an analog for modern Mars surface processes, but the atmospheric formation mechanism of Atacama nitrate and aerial deposition may not be analogous.
297-299 Nitrate on Mars, if all ancient and related to impact, would reflect processes other than simply sedimentation rates deposition, specifically aqueous transport processes present on a wetter Mars. In situ Mars data does not reveal either way whether the distribution is controlled by post depositional transport/leaching or simple deposition/accumulation.
83-84 this sentence does not relate well to the rest of the paragraph. Do you mean that these biofilms were discovered, or are present? Reword.
Figure 5, correlations not that strong, and sometimes controlled by 1 points as in 5a.
Fig. 6 & 7 may be a bit inaccessible to non-statistician.
383 – concentrations of nitrate and TOC tightly regulated by local sedimentation rates. But nitrate concentration may also be heavily influenced by post depositional processes such as leaching and transport, particularly under the scenario where all/most nitrate was deposited early in martian history when surface water was still present.
386-387 – if you subscribe to the idea that all nitrate on Mars is the product of low rate of atmospheric production and accumulation on the surface, then earlier in Mars history there would have been less nitrate present on the surface of Mars. Under this scenario nitrate concentration in the surface would be building up over the geologic timescale.
Reviewer 4 Report
The paper examines the nitrate, carbon, particle size, and total chemistry of 7 soils from N to S in Chile. Additionally, culturale organisms - both from the field and after the addition of water and/or nitrate were examined. The manuscript suggests this is relevant to Mars.
I can not support the publication of this manuscript, and suggest the authors rethink this whole thing. First, the sampling/experimental design: what of overarching importance can be learned from soil samples from 10-20cm? Soils a deep bodies, and the entirety of processes that impact them must be discussed in context of a depth integrated view. The authors seem unaware of the immense differences in the sources of the samples they examine from N to S (dust/aerosols in the N to likely (though not described) lithological material in the S. If they had examined total nitrate in the soil (and also included the ages of the soils) they would have provided a better story - but one that has already been approached and published by others. The correlation between Si/Al and nitrate is just that, but likely has little physical meaning except maybe for the aerosol/lithological source of the material.
Additionally, it rained just before this study, which likely redistributed NO3 and impacted the microbes. Earlier studies in the area had much less of a recnet impact on many features.
I think the climate/NO3 trends are uninterpretabe since one 10 cm increment was examined. Maybe the authors should focus entirely on the microbiology, which while sort of classic in approach (plating) does show some trends.
However, the geological context of each site needs to be given. Are they on alluvial fans, and if so, how old? What is the morphology of the soils? 20 cm tells us almost nothing about long term processes. There needs to be a rationale why Si/Al and NO3 have any theoretical menaing, and the concept of "sedimentation" needs to be defined - I do not know what it means in this context.
There is a LOT of very detailed geochemical, pedological, and biological work that has been laid out in this region. This study doesn't build on this, or seem to really take the step to match the detail of some of these works. It appears to be the result of a multi-hundred km drive from the N to S, with reconnaissance stops along the way.
Round 2
Reviewer 1 Report
The manuscript has been improved subject to my initial comments.
Author Response
We sincerely thank the reviewer for support.
Reviewer 3 Report
This revision is much improved. Minor revisions requested for:
l. 73 "Analagous processes can..." should be changed to "analagous processes could..." as there is no observational evidence that there is atmospherically derived nitrate currently being deposited on the surface of Mars. The Catling and Smith papers referenced here describ modeling studies which have not been ground truthed by observations.
l. 73-75. The mention of inferred nitrate/perchlorate ratios should be removed, unless there are plans to further discuss it. It doesn't add anything without further disccussion.
l. 75 and conclusion. Thermal shock due to impact heating would only affect nitrate formation on ancient Mars, not modern Mars. In the conclusion l. 481-485, impact shock heating is treated as if it is a modern process, which it is not. It only affects nitrate concentrations on early Mars.
Author Response
Lines 73-75 and Conclusions have been revised accordingly by changing “can” to “could”, removing the NO3/ClO4, and clearly discussing impact shock heating as an ancient Mars event.